# Olfactory Detection of Toluene by Detection Rats for Potential Screening of Lung Cancer

**DOI:** 10.3390/s21092967

**Published:** 2021-04-23

**Authors:** Yunkwang Oh, Oh-Seok Kwon, Sun-Seek Min, Yong-Beom Shin, Min-Kyu Oh, Moonil Kim

**Affiliations:** 1Bionanotechnology Research Center, Korea Research Institute of Bioscience and Biotechnology (KRIBB), 125 Gwahang-ro, Yuseong-gu, Daejeon 34141, Korea; oyk0213@kribb.re.kr (Y.O.); ybshin@kribb.re.kr (Y.-B.S.); 2Department of Chemical and Biological Engineering, Korea University, 145 Anam-ro, Sungbuk-gu, Seoul 02841, Korea; 3Infectious Disease Research Center, Korea Research Institute of Bioscience and Biotechnology (KRIBB), 125 Gwahang-ro, Yuseong-gu, Daejeon 34141, Korea; oskwon79@kribb.re.kr; 4Department of Physiology and Biophysics, Eulji University School of Medicine, 77 Gyeryong-ro, Jung-gu, Daejeon 34824, Korea; ssmin@eulji.ac.kr; 5KRIBB School, Korea University of Science and Technology (UST), 217 Gajeong-ro, Yuseong-gu, Daejeon 34113, Korea

**Keywords:** animal nose, odor detection, detection rat, olfactory behavior

## Abstract

Early detection is critical to successfully eradicating a variety of cancers, so the development of a new cancer primary screening system is essential. Herein, we report an animal nose sensor system for the potential primary screening of lung cancer. To establish this, we developed an odor discrimination training device based on operant conditioning paradigms for detection of toluene, an odor indicator component of lung cancer. The rats (N = 15) were trained to jump onto a floating ledge in response to toluene-spiked breath samples. Twelve rats among 15 trained rats reached performance criterion in 12 consecutive successful tests within a given set, or over 12 sets, with a success rate of over 90%. Through a total of 1934 tests, the trained rats (N = 3) showed excellent performance for toluene detection with 82% accuracy, 83% sensitivity, 81% specificity, 80% positive predictive value (PPV) and 83% negative predictive value (NPV). The animals also acquired considerable performance for odor discrimination even in rigorous tests, validating odor specificity. Since environmental and long-term stability are important factors that can influence the sensing results, the performance of the trained rats was studied under specified temperature (20, 25, and 30 °C) and humidity (30%, 45%, and 60% RH) conditions, and monitored over a period of 45 days. At given conditions of temperature and humidity, the animal sensors showed an average accuracy within a deviation range of ±10%, indicating the excellent environmental stability of the detection rats. Surprisingly, the trained rats did not differ in retention of last odor discrimination when tested 45 days after training, denoting that the rats’ memory for trained odor is still available over a long period of time. When taken together, these results indicate that our odor discrimination training system can be useful for non-invasive breath testing and potential primary screening of lung cancer.

## 1. Introduction

Some animals with excellent olfactory discrimination capabilities (e.g., dogs, mice, bees, etc.) can be conditioned to identify target volatiles [1,2]. The conditioned animal can quickly recognize the unique odor of volatile organic compounds, and exhibit particular signal behaviors, acting as an animal nose sensor. Animal nose sensors have mostly been applied in the field of drug detection [3,4] and mine/explosives detection [5,6,7] and, recently, in some medical applications, such as cancer detection [8,9,10] and tuberculosis detection [11,12]. With regard to cancer diagnosis, early diagnosis is especially important for successful treatment of cancer [13]. In line with this, various molecular diagnosis [14,15], immunodiagnosis [16,17] and imaging technologies [18,19] have been developed for early diagnosis of cancer. Recently, analysis of volatile organic compounds (VOCs) released from human exhaled breath are becoming an important early diagnostic method that can be used for health evaluation [20,21]. Above all, breath testing has the advantage of eliminating the invasive sample collection required to test for disease markers present in blood or tissue. The disease-specific VOCs contained in the patient’s breath can be used as diagnostic indicators. Animal nose sensors, which utilize the animals’ olfactory sense to discriminate disease-specific odors, have all of these non-invasive and early detection benefits. There is growing interest in detection animals in the areas of forensic science, land security, and disease diagnosis [1,22].

Animal sensors can quickly, accurately, and selectively recognize numerous odor information contained in a patient’s breath through neural network. In contrast, electronic nose (e-nose) sensors that mimic the olfactory system of animals have accuracy problems because the amount of data that artificial neural network can process is limited. The e-nose has another limitation in that it cannot specifically respond to similar odors due to its poor selectivity. In this respect, animal sensors are advantageous over artificial olfaction. When it comes to animal nose sensor-based breath testing, it began with the fact that certain metabolites accumulate in certain cells or tissues, and disease-specific odors are associated with the accumulation of metabolites that are not excreted in mutant or mutagenic diseases. [23]. The use of sensitive animal nose sensors can be an interesting strategy for early monitoring of disease by detecting gaseous biomarkers present at low concentrations of odor components in a patient’s breath. Existing disease diagnosis studies based on animals have mostly been performed using detection dogs. In 1989, Williams and Pembroke first published in Lancet the results of an odor detection dog study for melanoma as a proof-of-concept [24]. Cornu et al. reported detection of prostate cancer using conditioned dogs, showing 91% sensitivity by detecting 30 out of 33 urine samples from prostate cancer patients and 91% specificity by detecting 30 out of 33 urine samples from normal patients [25]. Detection of rectal cancer using detection dogs was reported by Sonoda et al., showing 97% sensitivity and 99% specificity for olfactory detection of rectal cancer from fecal samples [26]. In addition to cancers, the detection of Clostridium difficile, which releases harmful toxins, was carried out using the trained dogs [27]. Although detection dogs are an attractive technique for recognizing specific odors, they have several limitations in terms of training efficiency and cost-effectiveness. For example, there is the problem of handler dependency [28,29], in which the dog reacts to the handler’s behavior rather than to the smell when the dog is performing detection. Also, when doing large-scale space training, gasping in dogs due to extreme physical exertion, results in a problem that significantly reduces the ability to detect [30]. These can be factors that deteriorate the detection performance of animals. In addition, the drawbacks of detection dogs include enormous burdens of time, space, and cost required for training the animals [31].

Rodents, with excellent olfactory sensitivity and odor discrimination ability, have many advantages as detection animals. Weetjens et al. reported that detection rats showed high sensitivity and negative predictive value when a patient’s sputum was tested for tuberculosis using a trained rodent [12]. The time and cost needed for training and handling animals are very low due to their small size and light weight [1,12]. Also, since animal training is performed using a small indoor device, reward or punishment is immediately provided according to the animal’s signal behavior, so that a delay in learning due to a timing error for providing reinforcement can be reduced [32]. In addition, detection dogs need more human resources to increase the number of animals, but detection rats need only increases in the number of small and inexpensive training devices to expand the number of animals. Accordingly, rodents can be simultaneously trained to detect specific odors in large groups of animals, so it is possible to obtain a sufficient amount of data for statistical analysis [33]. Based on the advantages of using rodents as odor detectors, in the current study, we developed a rat training system for detection of toluene in spiked breath samples. Trained animals showed high detection performance for a potential odor indicator component for lung cancer. The obtained results indicate that the animal biosensor system can potentially be useful for disease monitoring or primary screening in the lungs and beyond.

## 2. Materials and Methods

### 2.1. Animals and Odor

Male rats (Wistar, over 4 weeks, Samtako Bio Korea, Kyunggi, Korea) were used for odor detection training. All animals were maintained under normal conditions with a 12-h light-dark cycle and were individually housed in transparent plastic cages with conditions adjusted to 25 °C temperature and 40% relative humidity (RH). Prior to training, rats were food-restricted and maintained at approximately 85% of free-feeding body weight. Initial food restriction was accomplished gradually over 2 days prior to the first day of odor discrimination training. After training had been initiated, rats were fed once per day after the training and had free access to water throughout. During all training courses, individual rats were systematically rewarded or punished according to their odor-sensing behavior. The olfactory behavioral results reported here were collected from a total of 12 rats. All experiments were approved by the Eulji University Animal Care and Use Committee (EUIACUC) and were performed as directed. The odor used as target was toluene (C_6_H_5_CH_3_, purity > 99.8%, Sigma-Aldrich, St. Louis, MO, USA). The target odor was used as a positive stimulant followed by rewards.

### 2.2. Spiked Breath Sampling

Breath samples were collected from healthy adult males and females (N = 4) aged 25–50 years. The procedure for collecting breath samples was as follows. Each exhalation provider held breath for 2 s and then exhaled in a Tedlar bag. When the Tedlar bag was inflated by about 80%, the bag was locked by turning the stopcock attached to the bag so that the odor did not leak. As for spiked breath sampling, we followed the method previously reported [34]. Briefly, 0.2 µL of toluene solution was dropped onto a small square of Whatman filter paper (2 × 2 cm^2^); the toluene-loaded filter paper was placed into a 60 mL syringe. Twenty mL of sample A was combined with 40 mL of sample B to prepare the target odors, and normal exhaled breath consisting of 60 mL of sample B only was used as a control odor (Figure 1). This is a non-invasive sampling method that allows untrained researchers to take samples. The concentration of toluene gas was calculated as follows: After the complete evaporation of toluene drop (0.2 μL) in 60 mL syringe, the number of mg of toluene can be calculated to be 173.38 mg by using the density of toluene (0.8669 g/mL). The gas concentration of toluene can be converted to be 0.0147 mg/m^3^ unit through 3-step dilution [First dilute: 0.2 μL of toluene drop in syringe volume (60 mL); second dilute: gas mixture of 20 mL of first dilute and 40 mL of breath air; third dilute: 4 mL of second dilute injected into chamber (0.087 m^3^)]. The unit of gas concentration can be converted from mg/m^3^ to ppm according to the conversion factor [1 ppm (in air, at room temperature) = 3.76 mg/m^3^]. The resulting toluene gas has the concentration of 3.9 ppb.

### 2.3. Odor Discrimination Training Device

The odor discrimination training device is a rectangular parallelepiped shape made of acrylic with a size of 600 mm × 520 mm × 280 mm (length × width × height). The device is divided by a separation plate into two chambers, a detection chamber and a behavior chamber. The detection chamber has an odor injection hole connected to the odor delivery tube (3 mm in inner diameter) outside the device, 35 mm above the bottom. The tube is connected to two syringes through which the experimenter can provide target or control odors. A ventilating fan (100 mm in size) is installed in the upper center of the detection chamber connected to a duct hose and serves to discharge odors inside the device outside of the laboratory. The behavior chamber has a floating ledge installed on the inside of the front wall. A food hole (15 mm in diameter) to provide food pellets is formed 35 mm above the floating ledge.

### 2.4. Performance Measurement

Data analysis in odor discrimination performance was calculated by the following standard formulas for voice prediction, including the following sensitivity, specificity, accuracy, positive predictive value (PPV), and negative predictive value (NPV) (Table 1).

## 3. Results and Discussion

### 3.1. Odor Discrimination Training System

The animal odor discrimination training system is composed of 5 units, as follows: (1) injection unit for delivering odors into the chamber; (2) ventilation unit for removing odor components remaining in the chamber; (3) sensing unit for recording altering behavior of animals in response to odors; (4) reinforcement unit for making odor-reward association by using a go/no-go operant conditioning paradigm; (5) Punishment unit for weakening undesirable responses of animals by presenting something unpleasant after the response.

Figure 2a,b represent a schematic diagram showing the layout of the odor discrimination training system. There are two chambers separated by a plate on both sides of the training device. One is the detection chamber where odors are provided through the odor injection hole. The other is the behavior chamber where animals exhibit target behaviors in response to the injected odors, followed by rewards and punishments according to their resulting behaviors. The separation plate is removed as a signal for odor injection, and target or control odors are injected from the injection unit. Injection unit consists of an odor tube and syringe. The odors are delivered into the chamber through an odor injection hole connected to the odor tube attached to the syringe. The target odor is prepared from a spiked sample in which toluene is mixed with an exhaled breath, while the control odor is prepared from an unspiked breath. The animal sniffs through the odor injection hole and determines whether to perform a signaling behavior. If the animal senses the target odor, it will move to the behavior chamber, jump onto the floating ledge, and receive a food reward. Otherwise, if a control odor is provided, the animal’s target behavior is to stay in the detection chamber instead of moving to the behavior chamber.

Staying in the detection chamber in response to the target odor or jumping onto the floating ledge in the behavior chamber in response to control odor, is regarded as an erroneous behavior. In this case, an unpleasant stimulus such as a loud noise or stick-banging is provided as a punisher immediately after the undesirable behavior. The animals can use a myriad of behaviors to alert their experimenter, such as scratching the floor, standing with hind legs, flipping the body, pressing the lever, etc. The alerting behavior can be set differently according to the animal’s propensity and research objectives. The residual odors remaining in the chamber are discharged to outside the chamber through the operation of the ventilator installed on the ceiling of the chamber. The odor discrimination training device can be modified according to the size and behavior of the animal.

### 3.2. Animal Training

The animal training configuration consists of 7 stages, as follows: (1) the stage of placing rats in the detection chamber; (2) the stage of delivering odors into the detection chamber; (3) the stage of detecting the odor by rats; (4) the stage in which a rat moves into the behavior chamber; (5) the stage in which alerting behavior of the rat occurs in the behavior chamber; (6) the stage in which reward and punishment are provided to motivate the behavior of the animal; (7) the stage in which the rat returns to the detection chamber. The training stages were configured to return to the 1st stage after the 7th stage. All stages from stage 1 to stage 7 proceeded in succession. Figure 3 shows the 4 key steps for odor discrimination training.

To adapt to the condition prepared for training, rats were allowed to explore the detection chamber freely for 3 min before starting training. If rats do not move into the behavior chamber within 10 s after the target odor is provided, they are considered as showing an error response and punished, regardless of the outcome of the signaling behavior. Food pellets were provided with tweezers through holes in the food; plastic tweezers were used to prevent animals from biting metal and damaging their teeth. When a target behavior occurred due to the injected odor, the experimenter immediately used a clicker to provide a click sound and then immediately provided a food reward to the rats. This is to bridge the temporal gap between the behavior and the food reward. Namely, the clicker’s click sound acts as a conditioned reinforcer, playing a role in strengthening the association between target odor and target behavior. The total amount of daily food pellets as reinforcement consumed by animals during the training process can vary from course to course, as the target or control odors are not provided in a fixed ratio. After each test was completed, the inside of the chamber was thoroughly ventilated for 40 s using a ceiling-mounted ventilator. Each test took about 1 min, including ventilation. In general, it is known that the rodent’s attention to the odor cue reaches its peak at around 30 min. In the present study, rats were trained once a day for a set equivalent to 20 tests that took 30 min. This allowed the animal to maintain an optimal learning state in the training device.

To improve the reliability of the sensors, it is important to minimize the error response. Specifically, in the case of an animal nose sensor based on an animal’s olfactory ability, the frequency of error responses is likely to increase when the experimenter knows which sample is the target one [29]. In this study, all tests were performed as blind tests. Therefore, the experimenter did not know whether the odor sample had target or control odor, which blocked the experimenter from providing animals with clues about odor information. Odor contamination can also cause false alarms. Therefore, to prevent cross-contamination of odors, syringes were replaced with a new one for each set, and tubes for odor delivery were thoroughly cleaned and deodorized after a set of training was completed. The exhaled breath samples used in this study were collected using the methods described in Materials and Methods section.

### 3.3. Measurement of Odor Detection Performance

To assess the odor discrimination performance, 15 rats were trained using breath samples spiked with toluene, one of lung cancer-related VOCs. Twelve rats among 15 trained rats reached performance criterion in 12 consecutive successful tests within a given set, or over 12 sets, with a success rate of over 90%. Here, data for only 3 rats showing the best detection performance out of the 12 rats that were trained are reported. Unlike device-based sensors, a live animal-based biosensor, which detects a specific odor through olfactory learning of the animal, has variables of the animal itself, in addition to the experimental and environmental variables. For this reason, only 25–33% of a population of candidate drug (or explosive) detection dogs can pass the thorough training program. Something similar goes for the detection rats. That is why we selected the three best performing rats in this study. The rats (N = 3) required 38 sets of training (= 760 tests) to learn to discriminate between target and control odors or to make odor-reward association. All of the trained rats reached the performance criterion, in 12 consecutive successful tests within a given set or over 12 sets with a success rate of over 90%, in acquisition of olfactory discrimination between two odors. The lowest and highest number of sets that achieved the criterion for toluene detection were 30 (600 tests) and 54 (1080 tests), respectively. As shown Figure 4, the rats showed excellent performance, discriminating the toluene-spiked samples with 82% accuracy, 83% sensitivity, 81% specificity, 80% PPV, and 83% NPV through a total of 1934 tests. These results indicate that the rats were efficiently trained in the training device to acquire the odor discrimination.

In breath tests using exhaled samples, samples were collected with a mixture of various interfering substances. Therefore, we attempted to conduct a rigorous test using garlic and onion odors as interfering substances to investigate whether the trained rats could discriminate the target odor even among strong interfering substances. It was found that the rats were able to properly identify target odors even in the presence of disturbing odor components contained in the breath, with performance similar to that during non-rigorous tests (data not shown). The concentration of toluene in the spiked breath used in the current study was approximately 3.9 ppb. Given that the exhaled toluene level in lung cancer patients is known to be approximately 80–100 ppb [35], the level obtained from our study is approximately 10-fold lower than that observed in the breath of early lung cancer patients. Thus, this result may have clinical significance.

The great advantage of breath testing is non-invasive diagnosis for various diseases [36]. The most commonly adopted methods to detect various types of VOCs are gas chromatography/mass spectroscopy (GC/MS) [37], photoionization detectors (PID) [38] and semiconductor metal oxide (SMO) based gas sensors [39]. Sun et al. reported a mini gas chromatography (GC) photoionization detector (PID) system integrated with a micro GC column to detect VOCs biomarkers for lung cancer [40]. In that study, the PID sensor detected toluene at concentration level of 7 ppm. A chemical resistance type exhalation sensor based on semiconductor metal oxide (SMO) has been proposed for toluene detection, and its detection limit (LOD) was as low as 20 ppb [41]. The toluene concentration of the spike breaths used in the current study was about 3.9 ppb, which is relatively low. Thus, we expect the live animal-based biosensors to be used as a tool for exhalation monitoring.

### 3.4. Stability Test for Temperature and Humidity

Since detection rats are living organisms, they are more susceptible to environmental conditions than device-based sensors. In particular, environmental temperature and humidity are important factors that can influence the odor detection performance. To test the influences of temperature and humidity on the animal sensor responses, indoor air temperatures were maintained at 20 °C, 25 °C, and 30 °C, while relative humidity was set at 45% RH, and indoor humidity levels were kept at 30%, 45%, and 60%, while the temperature was set at 25 °C. As shown in Figure 5a,b, under controlled laboratory conditions, the target odor was discriminated by the trained rats with outstanding performance for olfactory detection of toluene. The performance of all rats measured under given temperature and humidity conditions was within a deviation range of ±10%, indicating that animal nose sensors showed excellent environmental stability. There were no obvious differences in accuracy, specificity or NPV among the three specified temperature levels. However, when the temperature was 20 °C, some increases in sensitivity and PPV were observed compared to the conditions at 25 °C and 30 °C.

Temperature determines the animal’s body temperature and can trigger changes in sensory systems, including the olfactory system, that affect its behavior [42]. The ambient temperature can modify the concentration of odorant compounds and affect the olfactory process. Olfactory regulation by temperature begins in the olfactory receptor organs [43]. Studies of olfactory behavior showed that animals subjected to temperature shift treatment were more sensitive than the control group and behaved as if they had smelled a lower concentration of odor. On the other hand, in lower temperature treatments, they behaved as if they had sniffed a higher concentration of odor [44]. This temperature-driven olfactory regulation may explain our obtained result of an increase in sensitivity and PPV at relatively low temperature of 20 °C.

### 3.5. Measurement of Long-Term Retention

The long-term retention of odor discrimination must be critically considered for practical use of the animal nose sensor. In an attempt to gain insight into the long-term stability of the animals, the odor detection performance of the trained rats was examined 45 days after the last discrimination learning set. As shown in Figure 6, when tested 45 days after training, the average accuracy, sensitivity, specificity, PPV, and NPV of the rats (N = 3) were observed to be 86%, 90%, 82%, 88%, and 81% through a total of 129 tests, respectively, indicating that they had good retention of odor information over a long period of time. The tested rats showed no significant differences in accuracy from the first set of odor discrimination to the last set. This result indicates that long-term memory for toluene appears to be unimpaired in all of the tested rats, thereby demonstrating that the odor discrimination system is useful to train rodents to learn to discriminate between target and non-target odors or to make odor-reward association.

The lifespan of olfactory neurons of rats is about 30–60 days, after which they are naturally replaced by new neurons [45]. When rats are consistently exposed to certain odor components and food-rewarded, changes occur in the neurons produced [46]. This means that the newly produced neurons will contain more receptors for the target odor. Specifically, when the hippocampus is continuously stimulated through iterative learning for odor discrimination, the hippocampus regards certain odors as critical information. The hippocampus transmits odor information to the olfactory cortex, and the learned odor information is firmly stored. This can explain the long-term retention of learned odor. In general, acquisition of olfactory discrimination is accompanied by synaptic plasticity in the olfactory cortex. Thus, it appears possible that synaptic mechanisms, such as long-term potentiation, can be maintained in the olfactory cortex of the tested rats.

## 4. Conclusions

In this study, we report an animal nose sensor system for detecting toluene, one of lung cancer-related VOCs. For this, we developed an odor discrimination training device for odor detection, which was designed with two major learning paradigms of reinforcement and punishment. This system exhibited excellent performance results for toluene detection and showed good stability at high and low temperature and humidity when tested using spiked breath samples. The trained rats also had good retention of olfactory discrimination over a long period of time. These results indicate that the animal nose sensor system can be a useful tool for non-invasive exhalation monitoring and primary early screening for diseases with gaseous biomarkers. In addition, the system has the potential to provide predictive information on prognosis as well as detection of disease-related odor components through breath analysis. The greatest advantage of animal sensors is the rapid, accurate, and sensitive detection of disease-specific VOCs in exhaled breath, whether identified or unidentified. In this study, we reported the detection of toluene in spiked breath samples due to absence of actual samples. Further study is thus essential to detect the presence, or absence, of a disease by measuring exhaled breath samples of real patients, and the results will be reported in next studies. Although this is an attractive technique for detecting disease-related odors, the system has a limitation in its need for manual control of each training course. In such a labor-intensive training system, it is difficult to train a large number of rats to perform tasks, and the olfactory learning of animals can be influenced by the handler. To solve these problems, it is necessary to develop an automated system in which all processes of rodent odor discrimination training are automatically controlled. Recently, Gao et al. reported an in vivo bioelectronic nose based on transgenic mice for specific odor detection [47]. The combination of genetic modification (GM) technology and animal nose detection technology is likely to yield a synergistic effect between both technologies. The clinical importance of early detection of cancer cannot be overemphasized. From this point of view, animal nose sensors capable of discriminating cancer-related odor components present in trace amounts in exhalation samples can be useful for potential screening of lung cancer in a non-invasive manner.

## Figures and Tables

**Figure 1 sensors-21-02967-f001:**
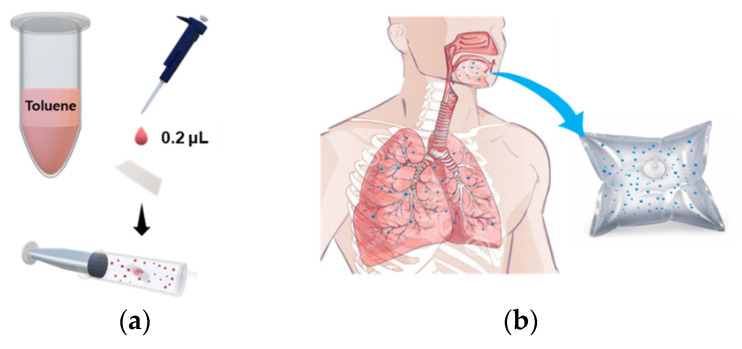
Schematic diagram of spiked breath sampling. (**a**) Sample preparation of toluene odor, (**b**) Exhaled breath from healthy individuals.

**Figure 2 sensors-21-02967-f002:**
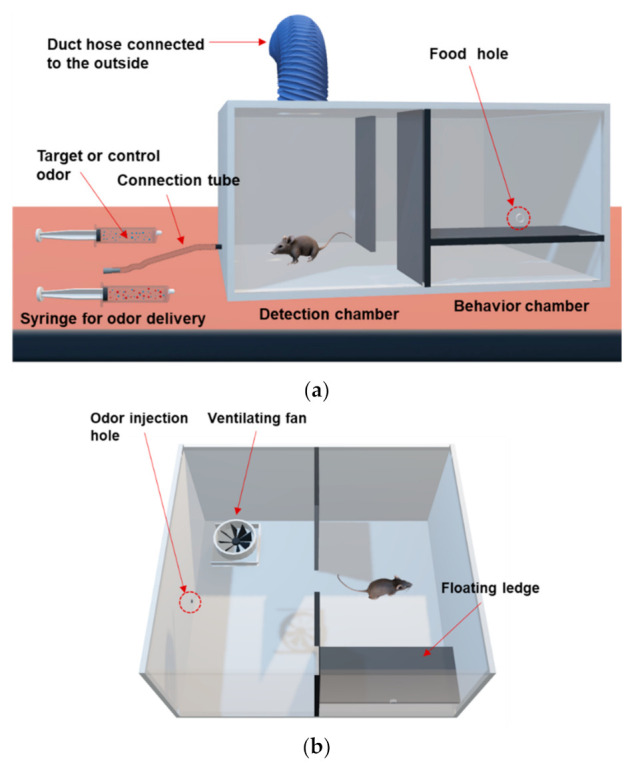
Schematic diagram of the odor discrimination training device. (**a**) 3D-front view, (**b**) 3D-bird’s eye view of the training apparatus.

**Figure 3 sensors-21-02967-f003:**
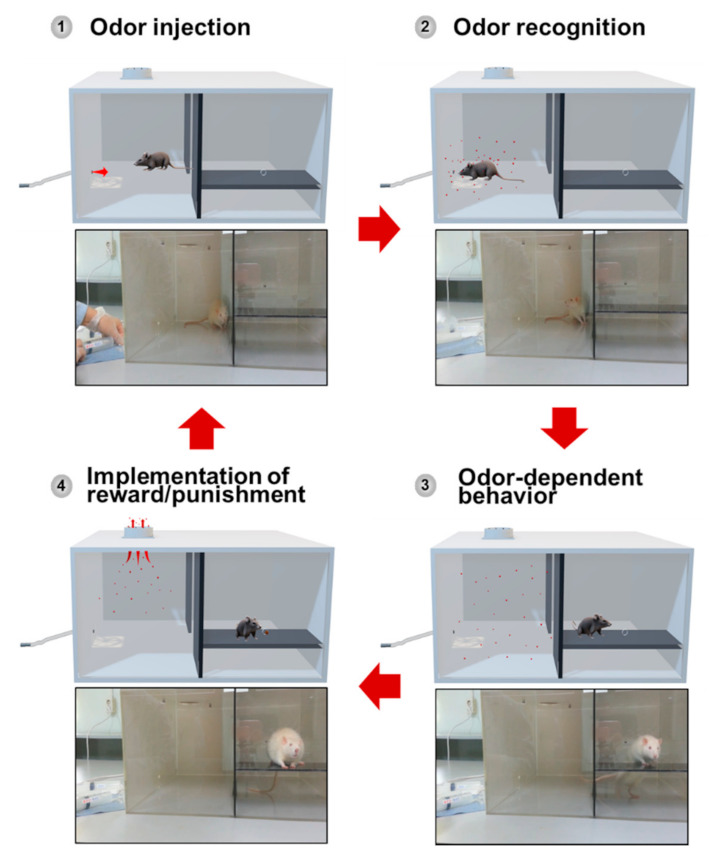
Four key steps for odor discrimination training. Step 1: odor injection. Step 2: odor recognition. Step 3: odor-dependent behavior. Step 4: implementation of reward/punishment.

**Figure 4 sensors-21-02967-f004:**
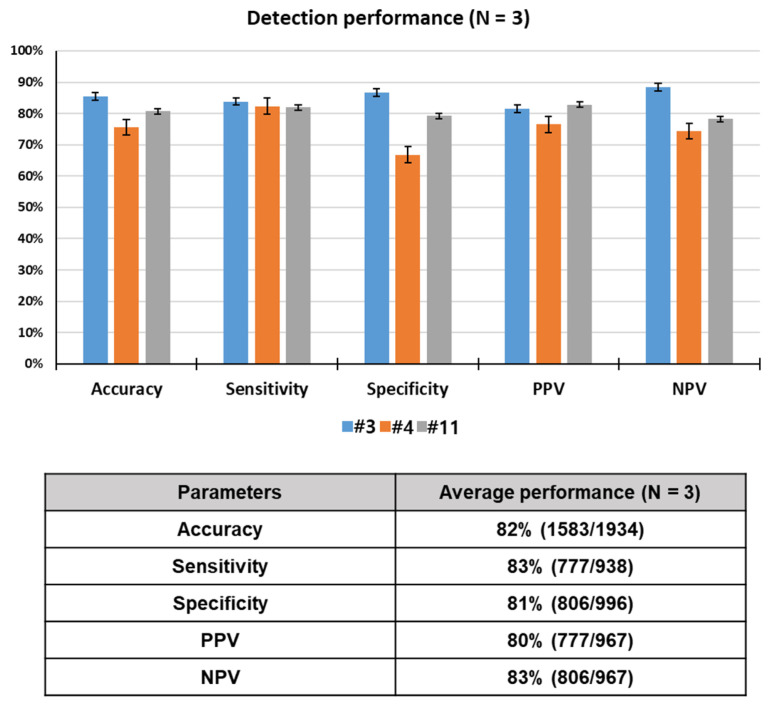
Rat performance detecting toluene in spiked breath through a total of 1934 tests.

**Figure 5 sensors-21-02967-f005:**
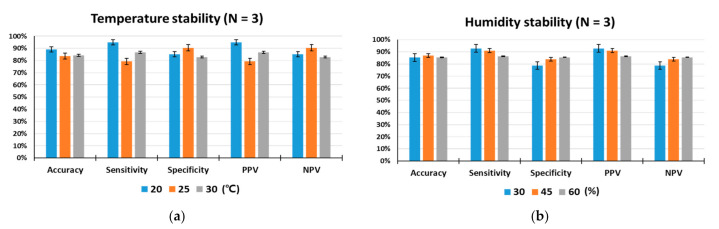
Analysis of environmental stability. (**a**) Temperature stability for a total of 1198 tests, (**b**) humidity stability for a total of 1247 tests. The performance of the trained rats (N = 3) was tested under different temperature (20, 25, and 30 °C) and humidity (30, 45, and 60% RH) conditions.

**Figure 6 sensors-21-02967-f006:**
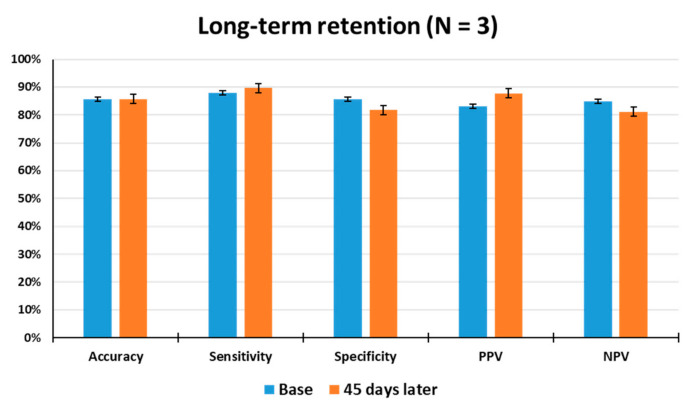
Measurement of long-term retention of olfactory discrimination. The odor detection performance of the trained rats (N = 3) was examined 45 days after the last discrimination training through a total of 129 tests. Base: Average test results for 1 week before long-term retention test.

**Table 1 sensors-21-02967-t001:** Data presentation in odor discrimination performance.

		Presence of Target Odor	
		Yes	No
Animal tests	Positive	a	b
	Negative	c	d

(1) Accuracy = (a + d)/(a + b + c + d) × 100 (%). (2) Sensitivity = a/(a + c) × 100 (%). (3) Specificity = d/(b + d) × 100 (%). (4) Positive Predictive value (PPV) = a/(a + b) × 100 (%). (5) Negative Predictive value (NPV) = d/(c + d) × 100 (%).

## Data Availability

The data presented in this study are available on request from the corresponding author (M.K.).

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
