# Peer review of "Olfactory Detection of Toluene by Detection Rats for Potential Screening of Lung Cancer"

_sensors, 2021, doi:10.3390/s21092967_

Round 1

Reviewer 1 Report

I am not an expert in mammalian olfaction or the use of mammalian noses in odor detection. Please take my comments in this context.

The system implemented seems quite complicated compared to the field utility of the design. I am guessing the intention is to place a rat near a human or human sample and based on the response, get a food reward or not? How do you plan to translate this system?

The training system seems to have low efficacy, if I am understanding this correctly. From your text, it appears that 3 of 15 rats learned the maze/reward system (?). This is 20%, which is low, I would think.

Specific comments

Pg 2. Metabolites cannot release odors.

The next sentence ‘The sensi-tive animal nose sensors to detect low concentrations of odor components in a patient's breath can be an interesting strategy for early monitoring of disease with gaseous bi-omarkers.’ Makes little grammatical sense. This is the second sentence in a row that needs language help.

Third sentence ‘Previous studies on the diagnosis based on detection animals have mainly been conducted using detection dogs.’ Also has issues.

*I am going to stop reviewing for language issues. This paper MUST be edited by a person with a strong command of English.

‘Cornu et al. reported detection of prostate cancer using conditioned dogs, which showed 91% sensitivity and 91% specificity by detecting 30 out of 33 urine samples from prostate cancer patients’

Without going to the paper, I can state this logic is incorrect. Yes 30/33 is about 91%. However, how many negative samples were screened for false positives? Revise sentence.

‘For example, there is a problem of handler dependency [28,29], in which dogs are likely to focus their attention on personnel.’ Cannot figure out what you mean.

‘Also, due to a large space training, the animal’s physical strength is exhausted, resulting in a problem of decreased attention to the odor cue [30].’ Not quite sure what you mean.

Para beginning ‘Rodents…’ What I think at this point is ‘why not just chemically screen for toluene?’ Why bother with inaccurate animal models? I was hoping you would make such a case in this para.

How does training get cheaper with a small animal? Do you need less muscular humans?

How about the negatives of the rodent model? Short lifespan, for example? Anxiety issues perhaps?

‘we developed the animal nose sensor system…’ correct me if I am wrong, but evo0lution developed the nose ‘system’. This language is apparently driving me crazy.

‘…for the lung cancer- specific odor component.’ Just one? Again, your language is seriously harming your message.

‘The obtained results indicate that our animal nose sensor system…’ how on earth is this yours?

2.1. ‘Odors’ = toluene? I was sincerely hoping you had a few.

2.3. Figure reference?

3.1 / 3.2. All of this is methods, unless the paper is a methods paper, which I am beginning to think it is.

‘If an animal senses target odor, it will move to the behavior chamber, jump onto the floating ledge, and receive food reward. Otherwise, if a control odor is provided, the animal's target behavior is to stay in the detection chamber instead of moving to the behavior chamber.

My chief concern is that the animal associates the platform with the reward, not the odor. Further, it may think, ‘if I move out of the behavior chamber, then back, jump on the platform, I should get a reward.’ It seems like you are asking for a series of things to occur: move out of the behavior chamber (find the door), move to the odor introduction area, sense the odor (+/-), move back through the door, move onto the platform, receive the food. Does this sound right? I may be quite wrong, but this seems crazy complicated for an odor sensing paradigm. Why not simply sense the odor, go to where the food is? Too simple?

And there is a click? Then why not just associate the click with food? Can the animal not decide to associate either? I don’t get it.

‘The target behavior can be set differently by changing the design of the sensing unit’ what does this mean?

Also, if a ‘sensing unit’ is a rat, please just say so.

Where is the ‘punisher’ targeted?

‘In case of a breath test using exhaled samples, the samples are collected with a mix-ture of various interfering substances. Therefore, we attempted to conduct a rigorous test using garlic and onion odors as interfering substances to investigate whether the trained rats could discriminate the target odor even in strong interfering substances.’ This is very important material. Youi need to state earlier in the paper that you controlled for odor specificity by including non-target odors, or similar.

Reviewer 2 Report

The authors present the results of the experiment in which they demonstrated that trained rats can detect a low concentration of toluene diluted in exhaled breath.
The results are interesting and presented as a possibility to use this method to detect early symptoms of lung cancer.

In my opinion, before acceptance of their manuscript, the authors should clarify some points.

The authors reviewed other research in which trained animals are used to detect odors. In my opinion in some applications use of animals has other advantages than only recognition of odors for example in case of searching for explosives or drugs as animals can enter various nooks and crannies. In the case of cancer detection use of animals seem to be more inconvenient than gas detector devices. In my opinion, the advantage of animals could be the ability to detect cancer by odor patterns in which there are no clear biomarker gases such as toluene. But it was not the case of the performed experiment since the aim was to detect artificially added toluene to the breath of healthy people. Maybe the authors have another view of this application and should elaborate on it.

The authors should review other methods of detection of toluene, for example using MOS of PID sensors, and compare them to their approach. In the proposed application they are competitive if their detection level is sufficient.

In figures 4-6 and the text of the manuscript the authors present percentages as measures of odor detection performance. It would be more meaningful if in the same figure and table the authors provide also the underlying numbers of tests in nominator and denominator e.g. 88% (22/25)).  

In figures 4-6 and text of the manuscript the authors present as results average values of the detection performance, when averaging is over 3 animals and over several performed experiments with each rat. Such an approach could be justified if in the intended application also measurements were performed in the same way so with 3 or more animals. More meaningful would be if authors present more detailed results where the variability between the performance of gas detection by individual animals could be estimated. Maybe the performance of the selected rats is almost the same but it is not clearly explained. My advice is to present the results for each animal. It could be done even in the same figures or if they would be too overloaded in separate figures.

In the experiment, the authors trained 15 rats and in the manuscript, they present results of toluene detection by only 3 animals with the best performance. The authors claim that the toluene detection performance by other animals is also satisfactory. However, I believe it is a bad practice to select only the best results. Maybe in the case of the intended application, the best-performing animals would be selected after training. But since the claim of satisfactory performance of all was expressed I think that also the results of other animals than only best 3 should be presented with similar details. Authors claim that the variability between animals is of the level of 10% so maybe they could present just the best and the worst case, as a visual demonstration. In my opinion, such a comparison is more important than the comparison of the influence of temperature and humidity presented by the authors, as these conditions could be more easily controlled than the differences between individuals.

Reviewer 3 Report

Comments to Authors

In general, the work is good to read, the structure of the work is clear. The research topic is very interesting and provide to important conclusion in the diagnosis of neoplastic diseases of the lungs. My comments below.

- Was the diet of healthy men and women the same prior to sampling? Diet has a direct impact on the volatile compounds profile in the exhaled air.

- The research lacks reference methods for the assessment of exhaled air samples. It seems that volatile substances in samples should be determined by gas chromatography. Which would allow to identify the main volatile compounds that, apart from Toluene, could influence the decisions of the rats.

- It seems that a good method of checking the VOCs profile would be to use an electronic nose to test the stability of the online profile?

- What other aromatic hydrocarbons (except toluene)  accompany lung cancer? I think that there may be more of such aromatic hydrocarbons in the profile?

- The use of an instrumental reference method in research (e.g. GC-MS or enose or others) is consistent with the topic of Sensors.

Round 2

Reviewer 1 Report

Changes are acceptable.

Author Response

Thanks!

Reviewer 2 Report

The authors improved their manuscript, however answers to some of the remarks they included only in the response letter and not in the main text. In my opinion, similar questions may have other readers so these issues should be addressed in the submitted paper. 

Author Response

Our responses to your valuable comments are included in the secondly revised manuscript on page 8 and 12. Thanks!

Reviewer 3 Report

Thanks for the answers and the proofreading of the manuscript.

Author Response

Thanks!